# Removal Characteristics of Effluent Organic Matter (EfOM) in Pharmaceutical Tailwater by a Combined Coagulation and UV/O₃ Process

**Jian Wang** [1,2], **Yonghui Song** [2,*], **Feng Qian** [2,3,*], **Cong Du** [2,3], **Huibin Yu** [2,3]
**and Liancheng Xiang** [2,3]

[1]  State Key Joint Laboratory of Environment Simulation and Pollution Control, School of Environment, Tsinghua University, Beijing 100084, China; jian-wan14@mails.tsinghua.edu.cn
[2]  State Key Laboratory of Environmental Criteria and Risk Assessment, Chinese Research Academy of Environmental Sciences, Beijing 100012, China; ducongducong@126.com (C.D.); yhbybx@163.com (H.Y.); xianglc@craes.org.cn (L.X.)
[3]  Department of Urban Water Environmental Research, Chinese Research Academy of Environmental Sciences, Beijing 100012, China
*  Correspondence: songyh@craes.org.cn (Y.S.); qianfeng@craes.org.cn (F.Q.); Tel.: +86-10-8492-4787 (Y.S.); +86-10-8491-7906 (F.Q.)

**Abstract:** A novel coagulation combined with UV/O₃ process was employed to remove the effluent organic matter (EfOM) from a biotreated pharmaceutical wastewater for harmlessness. The removal behavior of EfOM by UV/O₃ process was characterized by synchronous fluorescence spectroscopy (SFS) integrating two-dimensional correlation (2D-COS) and principal component analysis (PCA) technology. The highest dissolved organic carbon (DOC) and ratio of UV₂₅₄ and DOC (SUVA) removal efficiency reached 55.8% and 68.7% by coagulation-UV/O₃ process after 60 min oxidation, respectively. Five main components of pharmaceutical tail wastewater (PTW) were identified by SFS. Spectral analysis revealed that UV/O₃ was selective for the removal of different fluorescent components, especially fulvic acid-like fluorescent (FLF) component and humus-like fluorescent (HLF) component. Synchronous fluorescence/UV-visible two-dimensional correlation spectra analysis showed that the degradation of organic matter occurred sequentially in the order of HLF, FLF, microbial humus-like fluorescence component (MHLF), tryptophan-like fluorescent component (TRLF), tyrosine-like fluorescent component (TYLF). The UV/O₃ process removed 95.6% of HLF, 80.0% of FLF, 56.0% of TRLF, 50.8% of MHLF and 44.4% of TYLF. Therefore, the coagulation-UV/O₃ process was proven to be an attractive way to reduce the environmental risks of PTW.

**Keywords:** pharmaceutical wastewater; UV/O₃; coagulation; fluorescence spectrum; removal characteristics

## 1. Introduction

Pharmaceutical wastewater is one of the important sources for emerging pollutants such as hormones, antibiotics and non-biodegradable organic intermediates [1–3]. The pharmaceutical residue usually entered the aquatic environment via sewage, and even low concentration can impact the drinking water and human health [4]. Despite undergoing biological treatment, the pharmaceutical residue cannot be completely metabolized [2,5,6]. Some nonbiodegradable and toxic substances still exist in biologically treated effluent (pharmaceutical tail wastewater, PTW). Therefore, intensive treatment must be carried out to realize harmlessness and reduce environmental risks [7].

After biological treatment, the biodegradability of PTW was low and not suitable for continued biological treatment [3,7,8]. The BOD₅/COD (B/C) of PTW was only close to 0.1, meaning that the

organic matter from PTW was difficult to biodegrade. Therefore, the effect of biological treatment would be very poor in the further processing, physical chemistry and some advanced oxidation techniques should be considered. Among these, coagulation and UV/O$_3$ were the most two common technologies to remove specific organic pollutants from wastewater, such as contaminated groundwater, drinking water, industrial wastewater and landfill leachate [9–11]. Combined application of coagulation and UV/O$_3$ are suitable to treat PTW, since it has a certain removal effect on most organic matter ranging from low to high molecular weight. Coagulation treatment can effectively remove suspended particles, colloidal particles and dissolved organic matter (DOM) in sewage [12]. UV radiation can stimulate ozone oxidation to produce highly reactive hydroxyl radicals, which have the ability to oxidize and remove almost any organic contaminants [13]. However, studies on using UV/O$_3$, coagulation or combined coagulation-UV/O$_3$ to remove organic components of PTW have rarely been reported. Understanding the removal characteristics of effluent organic matter (EfOM) of PTW was an important basis for examining the effect of coagulation-UV/O$_3$ treatment. Therefore, it was necessary to find alternative characterization methods to evaluate the removal of EfOM during the oxidation process.

　　DOM, as the most important part of EfOM, can determine the coagulant, disinfection by-products, membrane fouling, type of microbial activity, and removal effect of contaminants [14,15]. The fluorescent component of DOM can be analyzed by fluorescence spectroscopy technology. The outstanding advantage of fluorescence spectroscopy is that it can quickly obtain measured DOM characteristic information with high sensitivity and has the characteristics of non-destructiveness and low cost [16,17]. Many fluorescent components in DOM can be analyzed and determined by combining fluorescence excitation-emitter matrix with parallel factors, self-organizing mapping algorithms or area integral methods [18]. Among them, synchronous fluorescence spectroscopy (SFS) was constituted by the measured fluorescence intensity signal and the corresponding excitation wavelength (or emission wavelength), based on simultaneous wavelength scanning of excitation and emission monochromators [19]. It has been widely used in the simultaneous analysis of heterogeneous mixtures, due to its advantages of simplified spectrum, reduced light scattering and improved selectivity. However, partial overlap of SFS wavelength can reduce its selectivity in DOM analysis. This problem can be solved through the combination of SFS and two-dimensional correlation (2D-COS) by extending peaks in the second dimension. Additionally, 2D-COS can be used as strong evidence for detecting correlations between features of dynamic spectra at two different wavelengths. It showed that combined SFS with 2D-COS can effectively describe the removal characteristics of DOM in PTW. Dynamic spectral changes are triggered by external disturbances, including various physical, chemical, and biological phenomena. The use of 2D-COS can explain the fundamental mechanisms of complex and heterogeneous material changes by enhancing the resolution of the spectrum and identifying the sequence of any subtle spectral changes in response to external disturbances [20,21].

　　Although the above methods have been widely used, there are few studies on the application of SFS technology and 2D-COS in the field of DOM structure in biotreated pharmaceutical wastewater by using coagulation-UV/O$_3$ combination method. The purpose of this paper is to (1) study the dynamic spectral changes of DOM in PTW after UV/O$_3$ oxidation using SFS technology, and (2) characterize the dynamic spectra and their relationship with different wavelengths by using 2D-COS combined with SFS technology, so as to reveal the degradation characteristics of different organic components in PTW.

## 2. Material and Methods

### 2.1. Pharmaceutical Tail Wastewater

　　The PTW was collected from the secondary settling tank of a pharmaceutical company wastewater treatment plant in Northeast China. The pharmaceutical company is a comprehensive enterprise, focusing on the production of chemosynthetic drugs and bio-fermentation drugs. The average daily production of pharmaceutical wastewater is 30,000 m$^3$. After being pretreated in a regulation tank, the wastewater was hydrolyzed and acidified in a two-stage hydrolysis acidification tank, and then

entered into a one-unit activated sludge reactor (UNITANK) for biological treatment. The water quality index of pharmaceutical wastewater before and after biotreatment was provided in Table 1. All samples were filtered through a 0.45 µm glass fiber membrane prior to measurement, except for turbidity.

**Table 1.** Characteristics of the pharmaceutical wastewater before and after biotreatment.

| Value | Value | |
| --- | --- | --- |
| | **Before Biotreatment** | **After Biotreatment** |
| pH | 5.3–6.1 | 6.8–7.2 |
| DOC (mg/L) | 739–892 | 77–126 |
| COD (mg/L) | 3331–5183 | 201–343 |
| SCOD | 2861–4658 | 180–277 |
| $NH_4^+$-N (mg/L) | 61–73 | 13–15 |
| B/C | 0.25–0.32 | 0.09–0.13 |
| $UV_{254}$ (/cm) | - | 0.906–1.31 |
| Turbidity (NTU) | 178–582 | 46–52 |
| Conductivity (mS/cm) | - | 14.95–15.57 |

## 2.2. Coagulation and UV/O₃ Treatment

The coagulation test was carried out in a six-unit agitator (ZR4-6, Zhongrun technology development Co., Ltd. Shenzhen, China). The 1000 mL pharmaceutical tail wastewater was placed in a cylindrical beaker with a volume of 1.5 L. The initial pH of the wastewater was adjusted by 0.1 mol/L of NaOH and HCl. In order to fully dissolve and react the added coagulant in the wastewater, the procedure of the agitator was set as follows: quick stirring for 2 min with a speed of 250 rpm, slow stirring for 10 min at a speed of 50 rpm. At last, the supernatant of treated wastewater was collected after 1 h of static treatment for the relevant water quality analysis and three-dimensional fluorescence analysis. Polyferric sulfate (($Fe_2(OH)_n(SO4)_{3-n/2}$) $_m$, PFS) was chosen as the coagulant in this study. The coagulant dosage is 0.4 g/L (according to the results of preliminary laboratory research), and the initial solution pH is 7.0.

Individual UV irradiation, $O_3$ oxidation and UV/$O_3$ oxidation experiments were processed in a closed double-layer cylindrical glass reactor with diameter of 8 cm, height of 13.5 cm and effective volume of 750 mL. The low-pressure UV lamp with the power of 15 W was used in this study to emit monochromatic light at emission spectra $\lambda$ = 254 nm. The UV lamp was placed in a 3.3 cm diameter quartz tube, which was placed in the center of the reactor. Both ozone generator (CFS20) and ozone concentration detector ($UV_{300}$) used in this study were purchased from Beijing Shanmei Shuimei Environmental Protection Technology Co., Ltd. (Beijing, China), and the ozone gas flow rate was controlled at 48–50 L/h to maintain the ozone concentration at around 10 mg/L (according to the results of preliminary laboratory research) during the $O_3$ oxidation and UV/$O_3$ treatment test. For comparison, the pH value of the biochemical treated pharmaceutical wastewater was adjusted to 7.0 by 0.1 mol/L $H_2SO_4$ and NaOH solution prior to individual UV irradiation, $O_3$ oxidation alone and UV/$O_3$ treatment, consistent with the pH value of the coagulation treatment. All the tests were carried out at room temperature (25 ± 2 °C).

## 2.3. Water Quality and Spectral Analysis

The turbidity was measured using a WGZ-1 turbidity meter from Xinrui Instrument Co., Ltd. (Shanghai, China). The dissolved organic carbon (DOC) was determined by Shimadzu TOC-VCPH analyzer (Shimadzu, Kyoto, Japan). Chemical oxygen demand (COD) was measured by spectrophotometry (DRB200, HACH, Loveland, CO, USA). OxiTop® system was used to determine $BOD_5$. The 254 nm UV absorbance ($UV_{254}$) was measured by UV-Vis spectrophotometer (UV-6100, METASH, Shanghai, China) with UV absorption spectrum in the range of 200–600 nm. The $NH_4^+$-N was determined by Nessler spectrophotometry. The analysis methods of the above conventional

indicators are all standard analytical methods published by the Ministry of Environmental Protection (HJ 535-2009). The SUVA value is defined as the ratio of the absorbance at 254 nm to the solution DOC concentration.

Synchronous fluorescence spectroscopy (SFS) was performed using a Hitachi fluorescence spectrophotometer (F-7000) from Hitachi, Tokyo, Japan. The response time of the spectrometer was set to 0.5 s and the photomultiplier tubes (PMT) voltage was 400 V. The wavelength range was from 260 nm to 550 nm by simultaneously scanning the excitation wavelength (ex) and emission wavelength (em). The passband was 0.2 nm and at the same time we kept the wavelength difference constant, $\Delta\lambda = \lambda_{em} - \lambda_{ex} = 18$ nm. The scanning speed was set to 240 nm/min. All spectral values were deducted from their respective program blanks. Fluorescence spectroscopy was usually performed within one day of sampling.

Two-dimensional correlation fluorescence spectroscopy (2D-COS) was analyzed by using the "2D-Shige" software developed by Kwansei-Gakuin University. Samples were taken at different time points from the UV/O$_3$ oxidation test, the sequence of time can be regarded as the specified external disturbance. Therefore, a matrix depending on sampling units at different points in time can be generated.

The synchronous correlation spectra can be calculated by the following formula [22]:

$$\phi(x_1, x_2) = \frac{1}{m-1}\Sigma_{j=1}^{m}I_j(x_1, t)I_j(x_2, t) \tag{1}$$

The asynchronous correlation spectrum is determined by the following formula [22]:

$$\varphi(x_1, x_2) = \frac{1}{m-1}\Sigma_{j=1}^{m}I_j(x_1, t)\Sigma_{k=1}^{m}N_{jk}I_j(x_2, t) \tag{2}$$

where $x$ represented an index variable (number of sampling points) of the synchronous fluorescence spectrum caused by disturbance variable $t$. $I_{(x, t)}$ was the analytical spectrum of $m$ evenly distributed on $t$ (between $T_{min}$ and $T_{max}$). $N_{jk}$ including the $j$ column and original element $k$ was called the discrete Hilbert-Noda transformation matrix and was defined as follows [22]:

$$N_{jk} = \{^{0}_{\frac{1}{\pi(k-j)}} \quad if = k;\ otherwise \tag{3}$$

The fluorescence intensity $\phi\ (x_1, x_2)$ of the synchronous fluorescence spectrum can represent a simultaneous or consistent change in the two independent spectral intensities determined by the perturbation variable $t$ between $T_{min}$ and $T_{max}$ at $x_1$ and $x_2$.

The spectral intensity of the asynchronous two-dimensional $\varphi\ (x_1, x_2)$ represented a continuous or continuous but inconsistent change in the spectral intensity measured at $x_1$ and $x_2$, respectively.

## 3. Results and Discussion

### 3.1. Changes in DOC and SUVA by Coagulation and UV/O$_3$ Process

DOC and SUVA are two important indicators for characterizing the organic matter in water treatment. The lower the SUVA value, the lower the risk of producing disinfection byproducts. As showed in Figure 1, after the coagulation pretreatment stage, the removal rate of DOC and SUVA reached 37.5% and 24.4%, respectively. After the oxidation stage, the DOC and SUVA was further reduced, the highest DOC and SUVA removal efficiency reached 55.8% and 68.7% by coagulation-UV/O$_3$ process after 60 min oxidation.

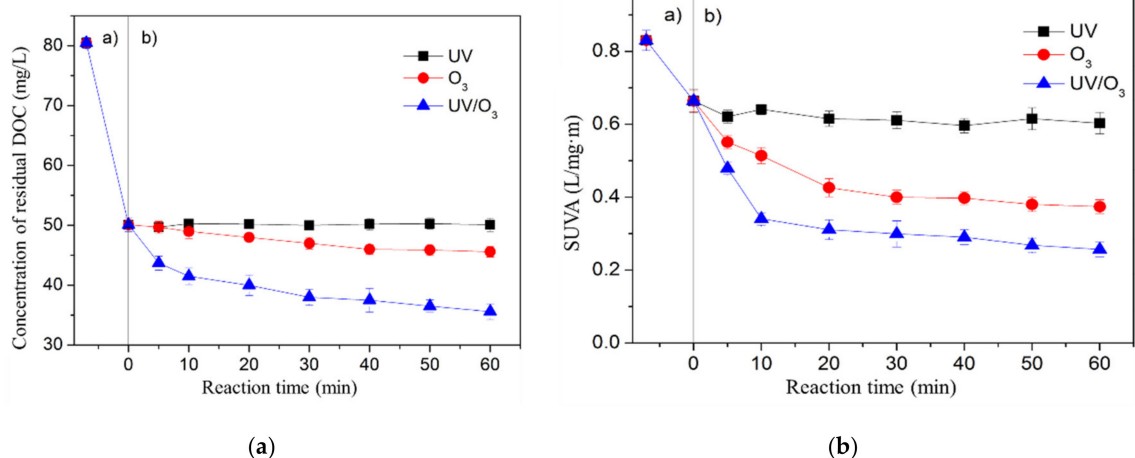

**Figure 1.** Effect of coagulation-UV, coagulation-$O_3$ and coagulation-UV/$O_3$ process on dissolved organic carbon (DOC) in pharmaceutical tail wastewater (PTW) (**a**: coagulation stage; **b**: oxidation stage).

The DOC concentration slightly changed during individual $O_3$ oxidation, indicating that the $O_3$ treatment mainly changed the structure of the organic matter by direct oxidation to form intermediate products instead of mineralizing organics into $CO_2$ and $H_2O$. Therefore, the 30.5% of SUVA was reduced by $O_3$ oxidation but only 5% of DOC was removed at the same time. In comparison, the ·OH formed by UV excitation in $O_3$ oxidation enhanced the degradation and mineralization of the active compound, thereby UV/$O_3$ achieved better DOC removal efficiency than $O_3$ oxidation.

Since the removal of organic matter was not achieved by UV irradiation alone, the effectiveness of UV/$O_3$ on PTW treatment demonstrated a synergistic effect of the combination of UV and $O_3$. In the UV/$O_3$ treatment process, the reduction in DOC content included direct and indirect effects. Direct effect was manifested in the photochemical mineralization of photosensitive substances (such as aromatic ring), and indirect effects represented the utilization of reactive oxygen species (OH), whose major production pathways require $O_2$ [23]. It can be seen in the first 20 min of the oxidation reaction that the SUVA value of the wastewater was rapidly decreased. Then, as time goes on, the decline rate of SUVA slowed down. At the end of the reaction (60 min), the removal rate of SUVA by $O_3$ or UV/$O_3$ reached 55% and 68.7%, respectively. These results indicate that organic components with UV absorption were easily oxidized by the $O_3$ or UV/$O_3$ process. The removal of SUVA (about 68.7%) was even higher than the removal rate of DOC. This indicates the preferential removal of aromatic luminescent chromophore or partially aromatic structure that can be converted to a non-UV absorbing compound by photochemical reaction. Many photochemical products were reported to have molecular weights as low as those of organic acids, alcohols, aldehydes, and inorganic carbons [24,25].

### 3.2. SFS Characteristics and Component Identification

### 3.2.1. SFS Characteristics and Fluorescent Component Removal

In order to reveal the removal mechanism of organics in PTW in the process of UV/$O_3$ treatment, the SFS characteristics of PTW taken at different time points during the oxidation treatment were analyzed. Three main peaks and two broad shoulders were shown in SFS (Figure 2a), including tyrosine-like fluorescent component (TYLF, $\lambda$ = 265–300 nm), tryptophan-like fluorescent component (TRLF, $\lambda$ = 300–360 nm) [26,27], microbial humus-like fluorescence component (MHLF, $\lambda$ = 360–420 nm), fulvic acid-like fluorescent component (FLF, $\lambda$ = 420–460 nm) and humus-like fluorescent component (HLF, $\lambda$ = 460–520 nm) [28]. It can be seen that the fluorescence intensity in the whole wavelength range decreased as the oxidation reaction proceeded. Similar to the fluorescence spectrum, the absolute

fluorescence loss of the solution during the reaction can be taken as a function of wavelength and compared over different irradiation oxidation times (Figure 2b).

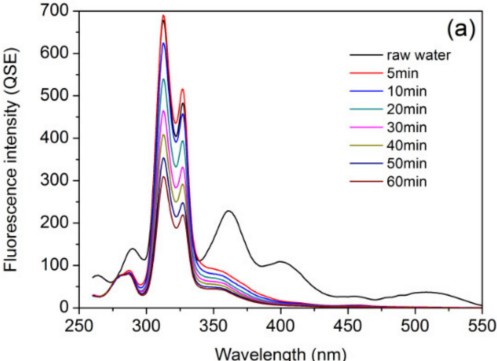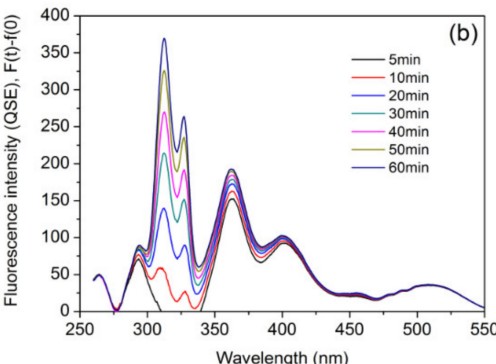

**Figure 2.** Changes in spectral responses of PTW with different irradiation times in UV/O$_3$ system: (**a**) synchronous fluorescence spectroscopy (SFS) ($\Delta\lambda$ = 18 nm) and (**b**) absolute fluorescence losses of excitation wavelength function.

As expected, the initial loss did not occur in the TRLF region, but in the TYLF, MHLF and the FLF component. The TYLF, MHLF and FLF components decreased rapidly and TRLF increased slightly in the first 5 min of UV/O$_3$ irradiation. This was likely to be due to the conversion of MHLF and FLF into TRLF during the irradiation oxidation.

### 3.2.2. SFS Component Identification and Principle Component Analysis

Principle component analysis (PCA) was carried out in this study to identify the component based on SFS at different times. The independent fluorescent components in PTW can be distinguished by cluster scores at different spectroscopic wavelength through PCA analysis. The Kaiser-Meyer-Olkin (KMO) test and Bartlett's test of sphericity (P) were two indexes to test the correlation between variables in the relevant array. In this study, the KMO value was 0.843, $p < 0.001$, indicating that SFS was well suited for PCA application [29]. After performing PCA analysis on the SFS of eight time points, two principal components (PC) were generated with the cumulative variance contribution rate of 99.69%, which can reflect most of the characteristics of the SFS (Figure 3). The scores curve of different wavelengths can show the characteristics of the spectral waveform of each principal component. Therefore, fluorescent groups that contributed variance to SFS signal can be identified [30]. PC1 with a variance of 98.8% showed three main peaks and four shoulders (Figure 3a). Same as the SFS characteristics, five fluorescent components were identified by wavelength at PC1, including TYLF (at 289 nm), TRLF (at 312 nm, 329 nm, 359 nm), MHLF (at 380 nm), FLF (at 450 nm) and HLF (at 470 nm). PC2 (variance of 0.89%) showed four main peaks and two shoulders (Figure 3b).

The main peak of TYLF at 292 nm exhibited a red-shift of 3 nm and its wavelength migration was longer than that in PC1, indicating that the polarity of the organic component was enhanced and the hydrophobicity was reduced. The MHLF peak was transferred to a shorter wavelength (360 nm) compared to PC1. The peak of FLF component had a red-shift of 5 nm at the wavelength of 455 nm. The TRLF component was identified at the wavelengths of 312 nm and 329 nm, no red-shift and blue-shift occurred compared PC1.

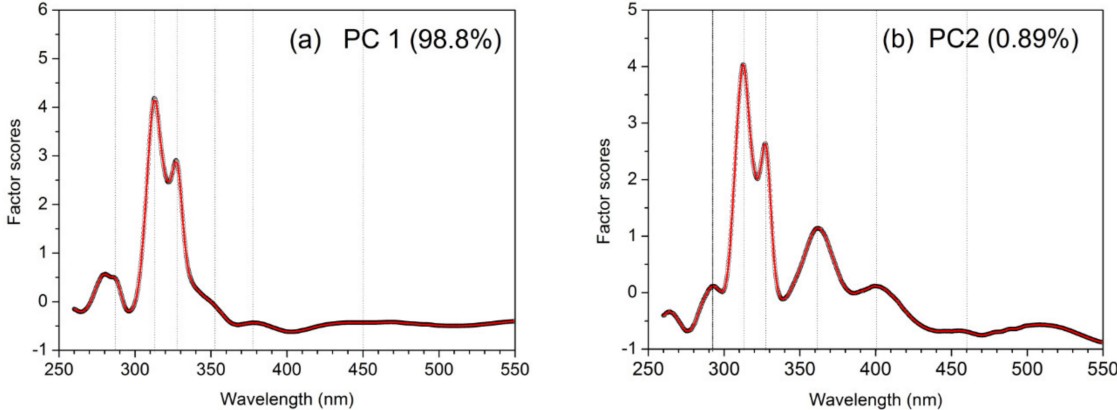

**Figure 3.** Scores plots of principle component 1 (PC1) (**a**) and principal component 2 (PC2) (**b**) for spectral wavelengths in UV/O$_3$ system.

### 3.2.3. Organic Component Spectral Area Integral

The SFS can be divided into five synchronous fluorescence regions based on excitation wavelength, which were corresponding to TYLF, TRLF, MHLF, FLF and HLF. The area integral of wavelength and fluorescence intensity can represent the relative abundance of homologous component. According to the area integral, the removal efficiency of fluorescence components can be studied. The distributions of the abundance of the DOM components were shown in Figure 4. It turned out that the DOC concentration showed a significant positive correlation to the variation of TYLF ($r = 0.940$, $p = 0.002$), TRLF ($r = 0.988$, $p = 0.0003$) and HLF ($r = 0.929$, $p = 0.003$), but a weak positive correlation to the MHLF ($r = 0.430$, $p = 0.335$) and FLF ($r = 0.499$, $p = 0.254$), which indicated that TYLF, TRLF and HLF could more accurately indicate DOC changes than MHLF and FLF. This was caused by the selective removal of fluorescent component in UV/O$_3$ irradiation oxidation process.

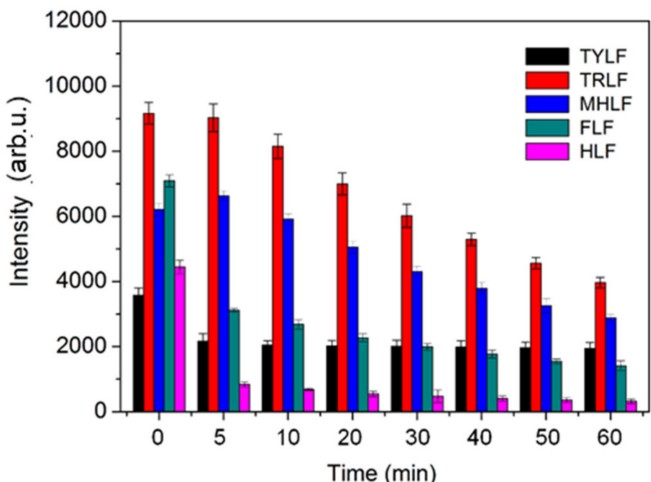

**Figure 4.** Distributions of the abundance of the dissolved organic matter (DOM) components in UV/O$_3$ system.

It can be seen in Figure 4 that the FLF, HLF and TYLF components decreased rapidly due to the UV/O$_3$ oxidation. Consequently, 82.2% of HLF, 40.4% of FLF and 40.0% of TYLF were removed in the first 5 min, indicating that the FLF, HLF and TYLF can be effectively removed by UV/O$_3$. This well explained the quick degradation of DOC and SUVA in the first reaction time, which was discussed before. However, in the subsequent reactions, the removal rate of FLF, HLF and TYLF gradually slowed down, and the TRLF and MHLF components began to decline. After 60 min of UV/O$_3$ oxidation,

the maximum removal efficiency was obtained by HLF component of 95.6%, followed by FLF of 80.0%, TRLF of 56.0%, MHLF of 50.8% and TYLF of 44.4%. This indicated that the UV/O$_3$ oxidation system could effectively remove the different fluorescent component in PTW, especially the HLF and FLF, while it took more time to completely remove the TYLF, TRLF and MHLF components.

*3.3. Synchronous Fluorescence/UV-Visible Two-Dimensional Correlation Spectra Analysis*

To reduce spectral overlap and reveal the DOM transformation process, 2D-COS was used to analyze two-dimensional fluorescence spectra [26]. Two-dimensional synchronous fluorescence spectrum of PTW was a symmetric spectrum about diagonals, in which there were four main self-peaks at wavelengths of 313 nm, 326 nm, 360 nm, and 400 nm (Figure 5a). The results are consistent with the PCA analysis. The self-peak represented the overall sensitivity of the corresponding spectral region after the change in spectral intensity as an external disturbance. During the reaction, the intensity of the self-peak at 313 nm and 326 nm was higher than that at 360 nm and 400 nm, indicating that the fluorescence intensity of the TYLF and TRLF component was clearly higher than that of the MHLF. The solid cross peaks indicated a positive correlation of fluorescence components at 313 nm, 326 nm, 360 nm, and 400 nm. Indirectly this confirmed the fluorescence components of DOM in PTW were synchronously removed in the process of the coagulation-UV/O$_3$ process. In the asynchronous fluorescence spectrum (Figure 5b), two negative cross peaks appeared at 289/313 nm and 289/326 nm, while five positive cross peaks were found at: 313/360, 313/400, 313/501, 326/360 and 326/400 nm. The positive or negative asynchronous crossover peak could provide information on the disturbance order of spectral band along the external variables. According to Nado rules, the change order of the spectral bands was: 501, 400, 360, 326, 313, 289 nm. This indirectly confirmed the removal characteristics of each component in the tail water treatment process of the synthetic pharmaceutical park, which had been discussed before.

The 2D-COS was used to study, in more detail, the characteristics of different absorption wavelengths of organic matter in the PTW during the oxidation process, the synchronous and asynchronous 2D-COS analysis using UV-visible absorption spectra with different oxidation time was shown in Figure 5c. Strong self-peaks at 225 nm and 255 nm and weak self-peak at 350 nm were observed in the synchronous 2D-UV-visible correlation spectrum and the intensity of the first two absorption bands at 225 nm and 255 nm decreased significantly with the irradiation time. This result was consistent with our previous observations of the overall trend of higher absorption losses at shorter wavelengths. In the asynchronous 2D-UV-visible correlation spectrum, three absorption bands of the oxidative chemical reaction were observed, and the wavelength ranges were: 200–240, 240–270, and 270–320 nm, respectively (Figure 5d). The sequence relationship of spectral variation characteristics with oxidation time can be derived: 270–320, 200–240, 240–270 nm. This indicated that the oxidation reaction first removed the macromolecular organic matter with a conjugated bond in the wavelength range of 270 nm–320 nm, and then removed the organic matter at 240–270 nm.

In order to further reveal the characteristics of organic matter removal and degradation, synchronous fluorescence/UV-visible two-dimensional correlation spectra were obtained. From the synchronous two-dimensional correlation map (Figure 5e), it can be found that the aromatic group at the wavelength of 255 nm in UV-visible spectrum was corresponding to the variation of TRLF, MHLF, FLF and HLF components at 313 nm, 326 nm, 360 nm and 400 nm in synchronous fluorescence. This indicated that the TRLF, MHLF, FLF and HLF components containing aromatic groups were degraded during the oxidation process. At the same time, the wavelength in the range of 400–500 nm in the UV-visible spectrum was opposite to the variation trend of the TRLF and MHLF components, which represented a humus-like substance with a high degree of humification and low lignin content that was generated after the oxidation reaction. Through analysis of asynchronous two-dimensional correlation map (Figure 5f), UV-visible spectrum at 285 nm (organic matter containing unsaturated conjugated double-bond structure) has a negative peak with *v1/v2* at 360 nm (FLF) and 400 nm (HLF) in synchronous fluorescence, respectively. It was shown that the organic compounds with unsaturated

conjugated double-bond structure in HLF and FLF was first degraded by the oxidation process. At the same time, the UV-visible spectrum at 255 nm showed a positive peak with *v1/v2* at 313 nm (TRLF) and 326 nm (MHLF) in the synchronous fluorescence, respectively, indicating the MHLF- and TRLF-containing aromatic groups were removed sequentially in subsequent oxidation. The spectral characteristics can explain that the degradation of organic matter occurred sequentially in the order of HLF→FLF→MHLF→TRLF.

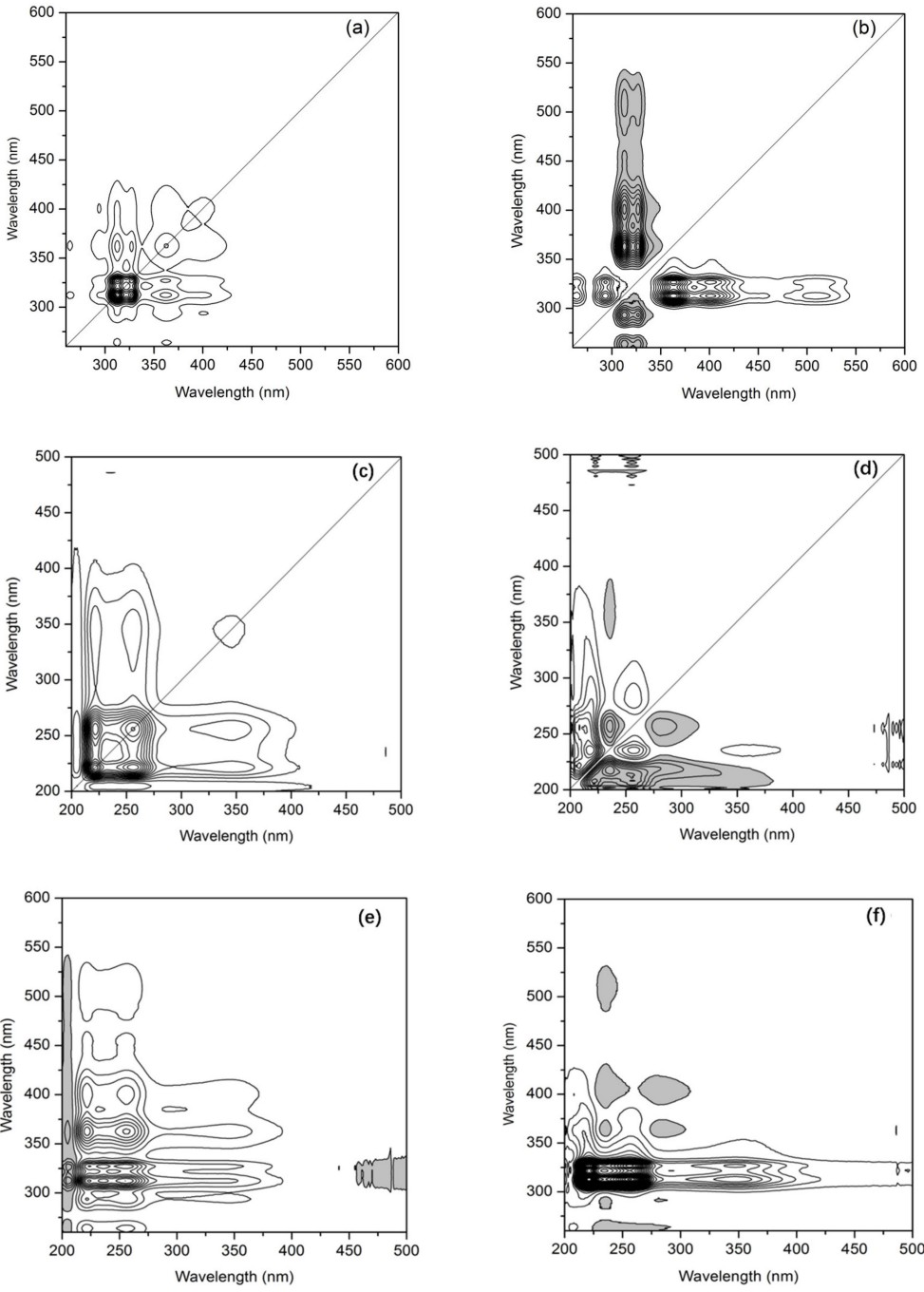

**Figure 5.** Two-dimensional- (2D)-correlation spectroscopy results of PTW: (**a**) synchronous fluorescence map, (**b**) asynchronous fluorescence map, (**c**) synchronous UV-visible map, (**d**) asynchronous UV-visible map, (**e**) synchronous fluorescence/UV-visible two-dimensional correlation map, (**f**) asynchronous fluorescence/UV-visible two-dimensional correlation map. The solid and the gray represent the positive and negative signs, respectively.

The organic matter in PTW can be effectively removed by coagulation-UV/$O_3$ pretreatment. After 60 min reaction, the removal rate of SUVA and DOC reached 68.7% and 55.8%, respectively. The UV can significantly enhance the mineralization of organic matters by $O_3$, therefore the treatment effect was significantly better than the $O_3$ oxidation alone. TRLF, TYLF, MHLF, FLF and HLF components were identified by the SFS combined with 2D-COS and PCA analysis. All the results consistently show that HLF and FLF were the main components of DOM in the tail water of synthetic pharmaceutical parks. UV/$O_3$ was selective for the removal of different fluorescent components, which can quickly remove FLF and HLF from the PTW and convert them into TRLF. The order of degradation of the different fluorescent components was HLF→FLF→MHLF→TRLF→TYLF. SFS combined with 2D-COS and PCA can quickly and effectively reveal the spectral dynamics of DOM in UV/$O_3$ treatment system and the removal and degradation characteristics of different organic components.

**Author Contributions:** Conceptualization, J.W. and F.Q.; methodology, J.W., H.Y. and C.D.; software, J.W., F.Q., C.D. and H.Y.; validation, Y.S. and L.X.; formal analysis, J.W., F.Q.; investigation, J.W., C.D., H.Y. and F.Q.; resources, Y.S.; data curation, Y.S.; writing—original draft preparation, J.W.; writing—review and editing, J.W. and F.Q.; visualization, J.W. and C.D.; supervision, Y.S. and L.X.; project administration, Y.S.; funding acquisition, Y.S. and F.Q. All authors have read and agreed to the published version of the manuscript.

**Funding:** This research was funded by National Major Program of Science and Technology for Water Pollution Control and Governance (Fund number, 2012ZX07202-005, 2018ZX07601-003, PR China). Research Project on Comprehensive Programme and Management Platform of Environmental Protection for the Main Stream of the Yangtze River and Typical Cities (2019-LHYJ01-0104).

**Conflicts of Interest:** The authors declare no conflict of interest.

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
