# Peer review of "Removal Characteristics of Effluent Organic Matter (EfOM) in Pharmaceutical Tailwater by a Combined Coagulation and UV/O3 Process"

_water, doi:10.3390/w12102773_

Round 1

Reviewer 1 Report

The presented publication concerns the important issue of the removal of pharmaceuticals from wastewater. However, certain assumptions of the work raise doubts that must be clarified.

0) Introduction could provide more information, why the proposed methods are competetive regarding biological treatment methods. 
1) Composition of the analyzed wastewater: there is no information about possible compounds or classes of pharmaceutical compounds present in the samples, at least approximate results of qualitative and quantitative analyzes are necessary.
2) The adopted criteria for assessing the effectiveness of the methods are of limited practical importance. In the treatment of industrial wastewater, the key is not to reduce DOM, but also to reduce the amount of toxic compounds. Will the method used not lead to the formation of toxic intermediates? Unfortunately, physicochemical oxidation often promotes such phenomena.
3) The authors used an advanced methodology for fluorescence analyzes. However, the question of quantifying the results remains. I have not noticed any mention of method calibration, calibration curves etc. The second question is on what basis were the wavelengths corresponding to each type of compound selected?
4) Table 1. The authors use the term "after treatment" but describe several variants of "treatment". To which do the results relate?
5) Could the authors discuss the issue of whether coagulation affects the concentration of pharmaceuticals, and not environmentally neutral compounds? Was the coagulant specific and selective?
6) The transition between coagulation and oxidation is unclear. Were they realized in different vessels? Was the coagulant in the reactor during the UV irradiation? Could there be desorption from its deposit?
7) Figure 4 only shows the results after UV/O3 treatment. The results for only UV and only O3 are missing.

Author Response

Dear Editor,

Thanks for your kind comments, I have prepared an answer sheet in details, please refer to the attachment. I am looking forward to hear from you soon.

Best Wishes,

Jian WANG

Reviewer 2 Report

In the abstract of this article was not described method and technology of coagulation.

The full text of article was contained describing of technology coagulation by addition of chemical coagulant and coagulant dosage.

 I would like suggest for author to calculate power expense in kwt/cubic meter of treatment pharmaceutical  wastewater, particularly for ozonation technologies.

Author Response

Dear Editor,

Thank you very much for your kind suggestion. That is really a good idea to calculate power expense of ozonation technologies. In the early stage of our research, we are looking for an effective method to treat PTW. In the next stage, we will carefully calculate the energy consumed in the treatment process, so as to treat the wastewater in the most effective and economical way. Otherwise, the focus of this paper is to analysis the change of fluorescence components in PTW during the treatment process. Therefore, the change of fluorescence components is introduced in detail in the abstract. The method and technology of coagulation is not introduced in the abstract, but there is an introduction in the main text.

Best Wishes

Jian WANG

Reviewer 3 Report

Dear Authors, you can find comments in attached file.

Author Response

Dear Editor,

Thanks for your kind comments, I have revised my manuscript in detail and answered your questions, please refer to the attachment. I am looking forward to hearing from you soon.

Best Wishes,

Jian WANG

Round 2

Reviewer 1 Report

Thank you the Authors for the kind responses. I am satisfied, however, I would suggest to include the content of the Response 4. into the manuscript, because it provide an important information, which are necessary to understantd the results properly.

Author Response

Dear reviewer,

Thank you very much for your kind comments, I think there must be more explanation added about Point 4. (1) According to the research of [26,27], The fluorescence diagram, you can see in the paper, is obtained by subtracting the program blank from the fluorescence spectrum obtained by actual scanning, so you will not noticed any mention of method calibration and calibration curves; (2) According to the research [26-28], the five typical peaks in the synchronous fluorescence spectra were determined as TYLF,TRLF,MHLF,FLF and HLF, with the corresponding wavelengths 265-300, 300-360, 360-420, 420-460, 460-520nm. You can see in the manuscript L192-L199. As follows,

Three main peaks and two broad shoulders were shown in SFS (Figure 2a), including tyrosine-like fluorescent component (TYLF, λ = 265–300 nm), tryptophan-like fluorescent component (TRLF, λ = 300–360 nm) [26,27], microbial humus-like fluorescence component (MHLF, λ = 360–420 nm), fulvic acid-like fluorescent component (FLF, λ = 420–460 nm) and humus-like fluorescent component (HLF, λ = 460–520 nm) [28]. It can be seen that the fluorescence intensity in the whole wavelength range decreased as the oxidation reaction proceed. Similar to the fluorescence spectrum, the absolute fluorescence loss of the solution during the reaction can be taken as a function of wavelength and compared over different irradiation oxidation times (Figure 2b).

26. Hur J. Microbial Changes in Selected Operational Descriptors of Dissolved Organic Matters From Various Sources in a Watershed. Water, Air, & Soil Pollution, 2010, 215(1-4):465-476.

27. Yu H, Song Y, Tu X, et al. Assessing removal efficiency of dissolved organic matter in wastewater treatment using fluorescence excitation emission matrices with parallel factor analysis and second derivative synchronous fluorescence. Bioresource Technology, 2013, 144:595-601.

28. Pan H, Yu H, Wang Y, et al. Investigating variations of fluorescent dissolved organic matter in wastewater treatment using synchronous fluorescence spectroscopy combined with principal component analysis and two-dimensional correlation. Environmental Technology, 2018, 39(19):2495-2502.

Best Wishes

JianWANG